# Comparative Analysis of the Response to Polyethylene Glycol-Simulated Drought Stress in Roots from Seedlings of “Modern” and “Ancient” Wheat Varieties

**DOI:** 10.3390/plants12030428

**Published:** 2023-01-17

**Authors:** Ilva Licaj, Maria Chiara Di Meo, Anna Fiorillo, Simone Samperna, Mauro Marra, Mariapina Rocco

**Affiliations:** 1Department of Science and Technology, University of Sannio, 82100 Benevento, Italy; 2Department of Biology, University of Tor Vergata, 00133 Rome, Italy

**Keywords:** drought stress, climate change, durum wheat, Svevo cultivar, Saragolla cultivar, proteomics

## Abstract

Durum wheat is widely cultivated in the Mediterranean, where it is the basis for the production of high added-value food derivatives such as pasta. In the next few years, the detrimental effects of global climate change will represent a serious challenge to crop yields. For durum wheat, the threat of climate change is worsened by the fact that cultivation relies on a few genetically uniform, elite varieties, better suited to intensive cultivation than “traditional” ones but less resistant to environmental stress. Hence, the renewed interest in “ancient” traditional varieties are expected to be more tolerant to environmental stress as a source of genetic resources to be exploited for the selection of useful agronomic traits such as drought tolerance. The aim of this study was to perform a comparative analysis of the effect and response of roots from the seedlings of two durum wheat cultivars: Svevo, a widely cultivated elite variety, and Saragolla, a traditional variety appreciated for its organoleptic characteristics, to Polyethylene glycol-simulated drought stress. The effect of water stress on root growth was analyzed and related to biochemical data such as hydrogen peroxide production, electrolyte leakage, membrane lipid peroxidation, proline synthesis, as well as to molecular data such as qRT-PCR analysis of drought responsive genes and proteomic analysis of changes in the protein repertoire of roots from the two cultivars.

## 1. Introduction

Wheat cultivation is fundamental for staple human and animal nutrition in vast areas of the world, particularly in the Mediterranean basin where it is the main source of carbohydrates for the population, as well as of revenues for farmers and food companies. Durum wheat (*Triticum turgidum* ssp. *durum*) is of high economic relevance since it is used for the production of semolina to make pasta. In Italy, the majority of durum wheat production is carried out in Southern regions, where traditional “ancient” cultivars providing high quality products but poorly suitable to intensive cultivation have been progressively substituted by a few high-yield, genetically uniform “modern” varieties. This fact represents a dangerous reduction in biodiversity. Drought has a markedly negative impact on crop productivity, especially in tropical and sub-tropical countries, but according model simulations of climate change predict that in the near future the impact of water deficit will also be widespread in temperate areas, such as the Mediterranean basin. Tolerance to drought is a complex trait, controlled by many genes and involving extensive cellular reprogramming, involving reactive oxygen species (ROS) production, osmolytes synthesis, metabolic adaptation, gene transcription and hormonal signaling [1,2,3]. ROS accumulation brings about membrane lipid peroxidation, macromolecule damage and photosynthesis inhibition [4]. Drought-tolerant species counteract ROS production by increasing enzymatic and non-enzymatic ROS scavenging activities [1,5,6]. Metabolic adaptation includes increased synthesis of sugars, polyols, secondary metabolites or amino acids and especially proline, due to its involvement in redox homeostasis [6,7], osmotic compensation [8], energy production and stress-related gene expression [9]. Transcriptional reprogramming involves the synthesis of stress-responsive proteins, such as Late Embryogenesis Abundant proteins (LEAs), including dehydrins (DHNs), chaperones, Heat Shock Proteins (HSPs) and mRNA-binding proteins [10]. Although different signaling molecules have been identified in the response to water stress [11], accumulated evidence demonstrates that ABA plays a major role in the induction and coordination of the adaptative response [12,13,14,15,16]. ABA transport and biosynthesis are increased in plant tissues in response to water deficiency [17], and ABA remodels plant growth during water stress by stimulating root development and inhibiting leaf expansion [18,19,20]. The effect of water deficiency is dependent on its duration and intensity, as well as on the plant growth stage [21]. Moreover, drought stress affects some tissues more severely than others. Fast responsive tissues undergo a greater diversity of anatomical, physiological and molecular changes compared to slower responsive tissues [22]. Roots are the organ where water deficit is first perceived and where morphological and molecular changes that allow plant protection from drought are initiated. Information about the effects induced by drought in roots mostly focuses on morphological, anatomical, and physiological parameters [23,24], whereas biochemical as well as transcriptomic and proteomic data are limited. A deeper understanding of the molecular basis of the response to water stress is fundamental to improve tolerance of wheat varieties in order to preserve productivity and quality, as well as to limit water consumption. Selection and genome characterization of drought–tolerant wheat varieties are crucial to identify peculiar traits to be used as markers in breeding programs for the production of water stress-ameliorated species. Currently, exploitation of the biodiversity of traditional and local varieties or landraces can provide a major contribution with the application of high-throughput “omics” technologies such as screening. In fact, in the last few years, the omic approach has successfully been used to perform comparative molecular studies leading to the identification of several stress-related genes as potential markers of drought tolerance in *Triticum* species [25]. In this study, following the renewed attention to “ancient” durum wheat cultivars from South Italy [26,27], we performed a comparative analysis of the response to Polyethylene glycol (PEG-6000)-simulated drought stress of roots from seedlings of the high-quality Saragolla traditional variety, renowned for its nutritional characteristics, compared to the Svevo elite variety, largely used for food industry applications. The effect of PEG-6000 treatments on root growth has been analyzed and correlated to biochemical data such as hydrogen peroxide production, electrolyte leakage, membrane lipid peroxidation and proline synthesis, as well as to molecular data such as qRT-PCR analysis of water-stress responsive genes and proteomic analysis of changes in the protein repertoire of roots from the two cultivars.

## 2. Results

### 2.1. Effect of Water Stress on Root Growth

Drought, similarly to other abiotic stresses, determines dramatic morphological, physiological and biochemical modifications that negatively impact crop growth and production [28]. Hence, the investigation of the effects of water stress on Saragolla and Svevo varieties began with an analysis of physiological and morphological parameters related to growth and fitness under stress conditions, such as water content and root-to-shoot ratio, respectively. To this purpose, seedlings of the two cultivars were grown in a climatic chamber at 24 °C and 50% humidity in hydroponic cultivation for 7 days. After this period, half of the seedlings of both cultivars were subjected to drought stress for 10 days, with the addition of PEG-6000 at 18% final concentration (corresponding to an osmotic potential of −1.0 Mpa) to the irrigated solution, whereas the remaining half of the seedlings was maintained in the control solution. In fact, PEG causes osmotic stress and is widely used to induce drought-like stress responses in plants. In Figure 1, the visual appearance of control seedlings or those subjected to 10 days of water stress is shown, whereas in Figure 2, the water content (Figure 2A) and the dry weight (Figure 2B) of roots after 2, 5 and 10 days of treatment is reported. Administration of 18% PEG-6000 brought about a reduction in the water content of roots in both varieties, which increased with the time of treatment. The water loss was significantly lower in the roots of Saragolla seedlings (4% as compared to 7% of Svevo seedlings, after 10 days of stress). Accordingly, PEG-6000 treatment also determined a reduction in the dry weight of roots that was lower in the Saragolla seedlings (36% as compared to 64% of Svevo seedlings after 10 days of stress). The relative water content (RWC) of roots and leaves after 10 days of stress was also determined. As reported in Table 1, the expected decrease in RWC in stressed roots and leaves was significantly lower in the Saragolla variety. In Table 1, the values of growth parameters measured after 10 days of drought treatment are reported. At the end of the treatment, the root-to-shoot ratio was increased in both cultivars and it was significantly higher in the Saragolla variety. The water content of leaves after 10 days of stress was also determined. As reported in Table 1, the water loss of leaves was markedly lower in the Saragolla variety (16%, compared to 48% of the Svevo variety). Overall, the above data indicate that drought impacted the growth of roots and the fitness of the Svevo variety more severely.

### 2.2. Effect of Water Stress on Biochemical Parameter

#### 2.2.1. Electrolyte Leakage

Since a consequence of water stress is the impairment of membrane functionality, with loss of the ability to control ion fluxes through the cell, the determination of electrolyte leakage from plant tissues is a reliable, although indirect, way to measure membrane insult [28]. Hence, relative electric conductivity (REC%) measures were performed in roots of Saragolla and Svevo seedlings subjected to 18% PEG-6000 treatment for 10 days. The results, reported in Figure 3, demonstrate that REC % values increased from 2 until 10 days of treatment in both varieties and were significantly higher in the Svevo cultivar, thereby indicating a reduced tolerance to membrane damage of this variety as compared to the Saragolla one.

#### 2.2.2. Malondialdehyde Production

Water stress determines a reduction in net photosynthesis, which in turn brings about thylakoid electron flow leakage to O_2_ by the Mehler reaction, thereby increasing ROS levels [1]. Unbalanced ROS generation causes oxidation of many cellular components, among which membrane lipids. Malondialdehyde (MDA) is produced from membrane-lipid oxidation by ROS, thereby being an indirect measure of membrane damage. Hence, to further estimate the detrimental effect of drought on membranes of roots from Svevo and Saragolla seedlings, MDA concentration in control and stressed roots was measured. Figure 4 shows that 18% PEG-6000 treatment highly induced MDA production in roots of the Svevo variety, whereas the ROS overproduction was comparatively lower in the Saragolla variety. This result was in accordance with those from electrolyte leakage determination and confirmed the higher susceptibility of the membranes of Svevo roots to water stress damage.

#### 2.2.3. Hydrogen Peroxide Production

ROS, including hydrogen peroxide, which is the most stable species, play a key role in the response of plants to diverse abiotic stress. ROS are, at the same time, signaling molecules necessary to mount the adaptative response, as well as toxic by-products of impaired metabolism that can threaten membrane and protein integrity [8]. It has been demonstrated that plants able to limit unbalanced ROS production during stress by means of detoxifying enzymes and endogenous reductants show a more effective adaptative response [4]. Consequently, we have determined the extent of hydrogen peroxide production in Saragolla and Svevo roots during osmotic stress treatment. The results reported in Figure 5 show that 18% PEG-6000 treatment increased H_2_O_2_ concentration in roots with a similar trend in both varieties: hydrogen peroxide concentrations were maximal after two days, then decreased significantly thereafter at 5 days. On the other hand, the increase in the control after 2 days of treatment was significantly higher in the Svevo variety. These results suggest that the antioxidant system of the Saragolla cultivar is more efficient in maintaining ROS homeostasis and that lower ROS levels determine a reduced MDA production and electrolyte leakage from membranes compared to those occurring in the Svevo cultivar.

#### 2.2.4. Proline Synthesis

Plants can accumulate intracellular free proline in response to abiotic stress, including drought [2,8,29]. This phenomenon is correlated to the decrease in water potential that plants bring about in order to facilitate water uptake, as well as to the protective role of proline towards membranes and proteins [30]. In particular, it has been demonstrated that proline accumulation during abiotic stress can counteract unbalanced ROS production [8]. Hence, we investigated whether the two cultivars responded to water stress challenges by increasing the proline content of root cells. The results reported in Figure 6 show that the proline content of the roots of both Saragolla and Svevo cultivars was enhanced after two days of stress administration and increased until the end of the treatment. The trend was quite similar in both cultivars, but the extent of accumulation was significantly higher in the Saragolla variety at every time point tested. It is conceivable that the higher capacity of accumulating proline of the Saragolla cultivar is positively related to the lower ROS content and reduced membrane damage, compared to the Svevo variety.

### 2.3. Effect of Water Stress on the Transcription of Stress-Related Genes

#### 2.3.1. WRKY Genes

The plant response to drought stress is quite complex and involves extensive gene expression reprogramming. Genes that encode for transcription factors (TFs) are key regulators of stress-responsive genes, potentially exploitable for the improvement of crop resistance to drought or other abiotic stress by marker-assisted selection or genetic transformation. Among the different families of TFs known to be involved in the tolerance to drought in model plants and wheat [31], WRKY is the largest one unique to plants [32], known to regulate not only the response to abiotic stress but also growth and development [33,34,35,36]. Hence, the effect of water stress treatment on the transcription of different *WRKY* genes known to be involved in the response to drought, namely *TaWRKY12*, *20*, *32*, *34* and *60*, was analyzed by qRT-PCR experiments. In fact, *TaWRKY12*, *TaWRKY32*, *TaWRKY34* and *TaWRKY60* have been found to be up-regulated in the roots or leaves of two varieties of bread wheat *(Triticum aestivum* L.) by different stress treatments, including drought [36], while over expression of *WRKY20* from *Glycine soya* or from wild soybean in alfalfa or *Arabidopsis*, respectively, has been demonstrated to enhance drought tolerance [37,38]. Results reported in Figure 7 show that after 5 days of stress administration, all of the selected *WRKY* genes were significantly up-regulated in both varieties and levels of up-regulation were markedly higher in the roots of the Saragolla variety. This finding also confirms that in durum wheat (*Triticum turgidum* ssp. *durum*) WRKY TFs play a pivotal role in the response to drought and suggests that *WRKY*s provide a relevant contribution to the improved drought tolerance of the Saragolla variety over that of the Svevo cultivar.

#### 2.3.2. DREB2, DHN3, WCOR410, and TaPUB1 Genes

In order to provide a more extensive evaluation of gene transcript modulation by osmotic stress, in addition to *WRKY*s, members of other classes of genes known to be involved in the expression of tolerance to drought were subjected to comparative qRT-PCR analysis. The transcription profile of *DREB2*, *DHN3*, *WCOR410* and *TaPUB1* genes in roots of Saragolla and Svevo seedlings after 5 days of stress administration was examined. Dehydration-responsive element binding proteins (*DREB*s) are a class of TFs modulated by different stress including drought, which in turn determine the expression of different tolerance-related genes [39]. In wheat, few *DREB*s have been functionally characterized and *DREB2* [40] is the member for which the most convincing evidence of modulation by drought has been obtained [39,41,42,43]. Dehydrins are hydrophilic proteins of the Late Embryogenesis Abundant II (LEA-II) group, which contrast desiccation-induced cellular damage occurring during development or caused by different stress, including drought [44,45,46]. *WCOR410*, a dehydrin originally reported to be accumulated in the plasma membrane upon cold stress [47], is highly induced in leaves of a tolerant wheat genotype upon water stress [2]. *TaPUB1* gene encodes an E3 ligase from *Triticum aestivum* (L.), whose constitutive expression in *Nicotiana benthamiana* enhances drought tolerance [48]. The results reported in Figure 8 show that after 5 days of stress, all of the selected drought-tolerance related genes were up-regulated in both varieties, particularly *DREB2*, whose relative transcription was about ten and five times the control values in the Saragolla and Svevo roots, respectively. For all genes, induction was higher in the Saragolla variety. Overall, the results relate well to biochemical data showing a higher tolerance to drought of the Saragolla variety over the Svevo one, confirming that transcriptional regulation plays a pivotal role in the expression of the improved drought tolerance of the Saragolla cultivar compared to that of the Svevo cultivar.

### 2.4. Proteomic Analysis

In order to analyze the differences in the protein repertoire of control and osmotically-stressed roots of Saragolla and Svevo seedlings, a proteomic approach based on 2D electrophoresis (2DE) quantification of resolved proteins and mass spectrometry identification of differentially represented species was undertaken. Proteins extracted from roots of Saragolla and Svevo seedlings grown under control conditions or treated with 18% PEG-6000 for 5 days were separated by 2DE in the pI range 4–7 (first dimension) and mass range 10–150 kDa (second dimension), followed by gel staining with colloidal Coomassie blue. Quantitative image analysis of stained gels allowed a comparison of the protein repertoire of stressed and control samples in order to determine statistically significant quantitative variations due to PEG-6000 treatment. Software-assisted statistical analysis revealed 30 protein spots that were differentially represented (fold change ≥2 or ≤0.5; *p* < 0.05) at least in one of the two cultivars. They were associated with 26 non-redundant protein species by mass-spectrometry analysis. The list of the identified proteins together with their quantitative variations is reported in Table 2, whereas the master gel from 2D analysis, together with spot numbers, is shown in Figure 9. Functional categorization according to Gene Ontology annotation and literature data allowed grouping of identified proteins into the following broad classes: abiotic stress, oxidative stress, energy/carbon metabolism and amino acid/nitrogen metabolism. The miscellaneous section included other identified proteins not belonging to the abovementioned functional groups.

#### 2.4.1. Abiotic Stress-Related Proteins

Four differentially expressed protein spots, corresponding to three different protein species related to abiotic stress response, were up-regulated by PEG-6000 treatment in both wheat varieties, namely dehydrin from *Triticum aestivum* (spot 9; DHN), annexin-P35 (spot 24; ANX-P35) and heat shock protein 70 (Hsp70) that was present as two distinct spots of different Mw and pI (spot 30: Mw = 95.1, pI = 5.34; spot 32: Mw = 95, pI = 4.98). Although DHNs are stress-related proteins primarily involved in protection against cellular damages induced by desiccation, members of this protein family are known to be induced by cold and drought in leaves and roots of different species, including wheat [42]. Interestingly, in a recent paper [49], it has been shown that in maize seedlings, osmotic stress or treatment with abscisic or salicylic acid, brought about induction of the maize *COR410* dehydrin gene, which was positively related to antioxidant enzymes activity and antioxidant molecules content increase, as well as to a decrease in ROS amount. In our conditions, up-regulation of DHN upon water stress treatment was higher in the Saragolla cultivar. Plant ANXs are a class of calcium-dependent phospholipid-binding proteins that play a role in protection from abiotic stress [50]. Different studies of transgenic ANXs overexpression have demonstrated their role in the mitigation of abiotic stress, particularly saline stress and drought [51,52]. The protective effect seems, at least in part, due to the capacity of ANXs to bring about a reduction in ROS levels, as it has been demonstrated in transgenic tobacco overexpressing the *TdANN12* ANX from *Triticum durum* [53]. In our conditions, up-regulation of ANX-P35 upon water stress treatment was almost double in the Saragolla cultivar compared to Svevo. In eukaryotic organisms, the conserved families of Hsp70 and Hsp90 chaperones intervene in response to different stresses to maintain protein homeostasis by facilitating protein folding [54]. Cellular levels of Hsp70 and Hsp90 increase primarily upon heat stress but are also enhanced by other abiotic stresses, including drought [55,56]. In our conditions, the amounts of Hsp70s were increased upon water stress treatment in both wheat cultivars, but to a significantly higher extent in the Saragolla one. Overall, from the above reported data, it appears that in the Saragolla variety a stronger expression of stress-related proteins occurred in response to drought treatment; in particular, this up-regulation involved proteins whose function may impact cellular ROS homeostasis.

#### 2.4.2. Oxidative Stress-Related Proteins

Six protein spots, corresponding to 5 different protein species, were identified as differentially expressed proteins at least in one of the two wheat varieties upon PEG-6000 treatment: glutathione S-transferase F3 (spot 1; F3GST), quinone reductase (spot 2; QR), mitochondrial superoxide dismutase (spot 5; Mn-SOD), peroxidase (spot 11 and 12; APX) and isoflavone reductase (spot 13; IFR). F3GST is a member of the Phi class of GSTs. GSTs constitute a ubiquitous protein family involved in plant growth and development. Ten classes can be distinguished, among which Tau and Phi are exclusive to plants and play a fundamental role in the detoxification of xenobiotics [57]. In addition, GSTs can act as GSH-dependent peroxidases, catalyzing the reduction in organic hydroperoxides to monohydroxy alcohols, therefore limiting the damage of oxidation [58]. Different studies have proven that GSTs in plants, besides development and metabolism, are involved in the stress response. In the wheat genome, more than 300 GST genes (*TaGST*) have been identified and by transcriptome analysis it has been determined that Tau and Phi isoforms respond to saline stress [59]. RNA-seq analysis revealed that they are highly expressed in roots where they are modulated by different abiotic stresses, including heat, cold and drought [60]. In this study, F3GST was three-times more up-regulated by water stress in the Saragolla variety than in Svevo. SODs are a small multigene family of proteins that protect from ROS toxicity by removing the most toxic superoxide (O_2_^-^), dismutating it to the less toxic species H_2_O_2_ and O_2_. In this way, SOD enzymes regulate the concentration of peroxides, signaling molecules associated with tolerance to different abiotic stresses [61]. SOD enzymes can be classified based on the metal cofactor they use, such as Cu/Zn, Fe or Mn. The Mn-SOD is predominantly found in mitochondria, which is a major site of ROS generation during stress [62]. An increase in SOD activity has been reported in various tolerant plants in response to different abiotic stress, whereas the transgenic overexpression of Mn-SOD genes enhanced the tolerance to salt in *Arabidopsis* and poplar [63,64], as well as to drought in *Arabidopsis* and rice [65,66]. Limited information is available for wheat Mn-SOD, but recently it has been demonstrated that overexpression in *Arabidopsis* of the Mn-SOD gene *TdMnSOD* from *Triticum durum*, enhanced the tolerance to salt, osmotic and oxidative stress [67]. In this study, up-regulation of Mn-SOD upon water stress treatment was moderately higher in the Saragolla cultivar. Two differential spots (11, 12) of diverse Mw and pI (spot 11: Mw = 41.4, pI = 6.67; spot 12: Mw = 41.3, pI = 6.63) were identified, corresponding to APX from wheat (*Triticum aestivum*), which is a member of the classical class III peroxidase subfamily exclusive of plants. Plant APXs are heme–containing enzymes which detoxify H_2_O_2_ produced by SOD, thereby contributing to ROS level homeostasis. Class III APXs are the most important antioxidant enzymes of the ascorbic acid/glutathione cycle, operating in chloroplasts and in the other major cellular compartments [68,69,70]. They are involved in a broad range of physiological processes, including growth, development, as well as response to abiotic and biotic stress [71]. Genome-wide analysis in wheat and *Aegilops tauschii* revealed that members of class III APXs are involved in the response to different biotic and abiotic stresses, including drought [72]. Characterization of the class III APX gene from wheat *TaPRX-2A* revealed that it was highly expressed in roots and induced by different stress, including osmotic PEG-6000 treatment. Overexpression of the *TaPRX-2A* gene in wheat determined higher tolerance to salt stress, together with increased levels of ROS-detoxifying enzymes as well as reduced content of malondialdehyde and ROS [73].

In our study, the APX up-regulation was significantly higher in the Saragolla cultivar. IFRs are key enzymes in the biosynthesis of isoflavonoid phytoalexins, occurring in leguminous plants in response to pathogen infection [74]. IFR-like proteins (IRL), which also belong to the IFR protein family, have been cloned from different species, including *Arabidopsis* [75], tobacco [76] and rice [77], and many of them have been implicated in the response to biotic or abiotic stress [58,75,76,77,78]. It has also been demonstrated that OsIRL from rice contrasts ROS increase during root development [77]. Information about IRLs from wheat is very poor; however, recently [79], using two near isogenic wheat lines differing in the expression of an *IRL* gene, it was demonstrated that IRL expression is related to heat stress tolerance, flavonoid content and total antioxidant capacity. In our conditions, IFR levels in the Saragolla cultivar were much more increased than in the Svevo variety, resulting among the most increased proteins by osmotic stress. Quinone reductase (QR) is part of the plant antioxidant system for ROS detoxification that includes also SOD, catalase (CAT) and APX. It has been shown that *Arabidopsis* plants overexpressing the QR gene *SmQR* from the halotolerant tree species *Salix matsudana* exhibited increased salt tolerance compared to wild type plants [80]. In our study, QR was up-regulated in both varieties upon osmotic stress, but to a higher extent in the Saragolla cultivar. Overall, from the above reported data, it can be concluded that in the Saragolla variety, a stronger expression of antioxidant proteins occurred in response to the osmotic stress treatment.

#### 2.4.3. Carbon Metabolism and Energy Production-Related Proteins

This group included 9 differentially expressed proteins, namely, malate dehydrogenase (spot 3; MDH), UDP-glucose 6-dehydrogenase3 (spot 15; UGDH), succynil CoA ligase (β chain) (spot 16; SCL), adenylosuccinate synthetase (spot 17; AdSS), enolase (spot 18; ENO), phytoene synthase (spot 19; PSy), cytosolic phosphoglycerate kinase (spot 20; PGK), phosphoglycerate mutase (spot 25; PGAM) and cytosolic aconitate hydratase (spot 29; ACON). All of them were up-regulated in both varieties upon water stress, except for UGDH and PGAM, which were down-regulated in the Saragolla variety, and ACON, which was not differentially expressed in the Saragolla variety. MDH is a fundamental enzyme of the tricarboxylic acid cycle (TCA), thereby playing a key role in the maintenance of cellular energy homeostasis. Interestingly, it has been reported that MDH levels increase in drought-tolerant barley [81] or wheat genotypes [82], whereas they decrease in susceptible ones. In our conditions, MDH showed the highest variation among all of the differentially expressed proteins, being strongly up-regulated in both varieties but especially in the Saragolla cultivar, with a fold change value of 35 compared to six of Svevo. This finding suggests that MDH plays a key role in the adaptation of wheat roots to drought in the Saragolla cultivar. MDH up-regulation is well related to SCL up-regulation. SCL participates to TCA, where it couples the hydrolysis of succinyl-CoA to the synthesis of ATP in the step of substrate-level phosphorylation. It was up-regulated in both varieties, but particularly in Svevo. Another differential protein involved in nucleotides production was AdSS, which catalyzes the synthesis of adenylosuccinate in the first step of the de novo biosynthesis of AMP from IMP. AdSS up-regulation was higher in the Svevo variety. Three enzymes of the glycolytic pathway for energy production were differentially expressed, namely ENO, PGK and PGAM. ENO catalyzes the conversion of 2-phosphoglycerate to phosphoenolpyruvate (PEP) and is considered a pacemaker enzyme, not only for the production of energy, but also of carbon skeletons for biosynthesis, since the product of the ENO reaction, PEP, represents a central metabolite in plant primary and secondary metabolism [83]. Although information about the involvement of ENO in the response to abiotic stress is very poor, the ZmEno1 isoform from maize was reported to play an important role during adaptation to anaerobiosis [84]. In our conditions, levels of ENO were strongly up-regulated by water stress in both varieties. Enhancement of ENO activity produces increased amounts of PEP that can be converted to pyruvate, thereby fueling respiration in the TCA cycle, as well as transformed to oxaloacetate by PEP carboxylase, to replenish the TCA cycle when organic acids are used for amino acid biosynthesis. Hence, ENO strong up-regulation is worth noting, since, together with the observed very high increase in MDH levels, it seems to confirm the central role of energy production pathways for the adaptation to drought in wheat roots. Interestingly, levels of the two enzymes that precede enolase in the energy-generating phase of glycolysis, PGK and PGAM, were also found to be up-regulated upon water stress only in the Svevo cultivar. In rice, it was demonstrated that a PGK isoform (OsPgk2) is induced by salinity stress, while its overexpression in tobacco improved salt tolerance end enhanced proline accumulation [85]. UGDH catalyzes the conversion of UDP-Glucose to UDP-glucuronic acid and is involved in the biosynthesis of hemicellulose and pectin [86]. Information about its role in response to stress is quite limited, but it was reported by proteomic analysis as an up-regulated protein in soybean seedlings subjected to flooding [87]. The amount of UGDH was increased in the Svevo variety by PEG treatment, whereas it was decreased in the Saragolla cultivar. ACON catalyzes the isomerization of citrate to isocitrate. Two isoforms of ACON have been detected in eukaryotic cells: mitochondrial, which is involved in the TCA cycle and cytosolic, which participates in different processes, such as cytosolic citrate metabolism [88,89,90] and glyoxylate cycle [91,92]. Furthermore, cytosolic ACONs have RNA binding activity, which in plants has been related to resistance to oxidative stress. In particular, in *Arabidopsis* and *Nicotiana benthamiana* it has been shown that transgenic reduction in cytosolic ACON activity increased resistance to methyl viologen [93]. In our conditions, ACON was up-regulated to the same extent in both varieties. PSy catalyzes the first committed step in the biosynthesis of carotenoids, which, besides functioning in photosynthetic light harvesting and protection, are also precursors of stress-elicited hormones such as abscisic acid (ABA). In bread wheat (*Triticum aestivum* L.), three PSy genes, *TaPSY1*, *2* and *3*, have been characterized and from expression analysis, it has been inferred that the *TaPSY3* genes are the most expressed in all tissues, as well as in response to heat and drought stress [94]. Interestingly, in rice (*Oryza sativa*), it has previously been demonstrated that the *OsPSY3* gene is up-regulated during increased ABA production upon salt stress and drought, especially in roots [95]. Under our conditions, PSy was strongly up-regulated in both varieties, a fact that suggests its basal involvement in the response to drought as a source of ABA in roots. Overall, the above data showed that levels of key enzymes of TCA cycle and glycolysis were significantly increased by drought in both cultivars, thus suggesting the pivotal role of energy generating metabolisms in the expression of tolerance in roots.

#### 2.4.4. Nitrogen and Amino acid Metabolism-Related Proteins

Eight differentially-expressed protein spots corresponding to six different nitrogen and amino acid metabolism-related proteins were identified. Glutamine synthetase isoform Gs1a (spot 10; GS), ferredoxin-nitrite reductase precursor (spot 14, spot 22 and spot 28; NiRspot 14: Mw = 52, pI = 6.24; spot 22: Mw = 57.1, pI = 5.83; spot 28: Mw = 65.7, pI = 6.85) and S-adenosylmethionine synthetase (spot 23; SAMS) were up-regulated by drought in both varieties; putative glycine decarboxylase subunit (spot 27; GDC), glutamine synthetase isoform Gsr1 (spot 7; GS) and cysteine synthase (spot 6; CSase) were up-regulated in the Svevo variety but not affected in the Saragolla cultivar. Two isoforms of cytosolic GS, namely Gs1a and GS1r, have been identified as differential proteins. GS is the key enzyme for ammonia assimilation in plants and its product is used for the biosynthesis of other amino acids, including proline [96], which is the most widespread osmolyte synthesized by plants in response to water stress. It has been shown that transgenic wheat (*Triticum aestivum* L.) expressing the C4 phosphoenolpyruvate carboxylase (PEPC) from maize is more tolerant to drought, a fact that is related to increased amounts of GS and proline, compared to untransformed wheat [97]. In our study, Gs1a was up-regulated to a much higher extent in the Saragolla variety compared to Svevo, while Gs1r was up-regulated in the Svevo cultivar and not affected in Saragolla. NiR is a plastidial enzyme that catalyzes the reduction of nitrite to ammonium, which is subsequently assimilated via GS and glutamate synthase (GOGAT) [98]. NiR activity is a control point in nitrate assimilation [99], hence it is conceivable that up-regulation of NiR (which was higher in the Svevo cultivar) is functional to increased nitrate assimilation and amino acid biosynthesis. In our study, this finding is well related to the observed increase in GS and of the other enzymes involved in amino acid metabolism. SAMS is a known drought-stress responsive enzyme which catalyzes the synthesis of methionine to provide methyls for the synthesis of osmolytes such as betaine, polyols and polyamines during water stress [100]. In our study, SAMS was strongly up-regulated in both varieties but particularly in the Svevo cultivar. CSase is involved in the last step of the synthesis of cysteine, a precursor of other sulfur-containing metabolites such as methionine and glutathione, which play important roles in plant development, metabolism and response to stresses. CSases are a gene family whose members have diverse tissue expression patterns or subcellular locations and may perform different functions [101]. Emerging evidence indicates that CSases are also involved in the response to abiotic stress, such as high-salt and heavy metals, and that their overexpression increases the tolerance to oxidative stress [101,102,103]. In our study, CSase was up-regulated in the Svevo cultivar and down-regulated in Saragolla. GDC is an enzyme complex known primarily for its pivotal role in photorespiration but also for participating in other metabolic pathways involving glycine and serine. Glycine and serine are precursors for betaine synthesis, which in different plant species, including wheat, is used as an osmolyte in response to salt and drought stress [104]. It has been demonstrated that GDC inhibitors in *Amaranthus tricolor* reduced the salt-induced accumulation of betaine [105], thereby indicating that the enzyme is, at least in part involved, in betaine accumulation. Overall, from the above data it can be inferred that drought stress brought about in both cultivars a significant up-regulation of enzymes involved in nitrogen assimilation and amino acid biosynthesis, which may function in cellular osmotic adjustment.

#### 2.4.5. Miscellaneous Proteins

Three proteins not classified in the above categories were identified, namely 14-3-3 protein (spot 8; 14-3-3) and ATP-dependent Clp protease ATP-binding subunit (spot 31; Clp P), which were up-regulated in both varieties and probable voltage-gated potassium channel subunit beta (spot 4; VGKC), which was up-regulated in the Svevo variety and down-regulated in Saragolla. Among this heterogeneous group of proteins, it is worth noting the identification of a 14-3-3 protein. In fact, 14-3-3s are a family of regulatory proteins widespread in eukaryotes that are involved in different cellular and physiological processes [106]. In plants, besides regulating ion transport, carbon and nitrogen metabolism, gene expression and signaling [107,108], they also participate in the response to diverse abiotic and biotic stresses [109,110], including drought, as has been demonstrated by the overexpression of the λ isoform of *Arabidopsis* in cotton [111] and the maize ZmGF14-6 isoform in rice [112]. In wheat, information about 14-3-3 functions is still quite poor, but recently it has been demonstrated that the 14-3-3 TaGF14b is up-regulated by drought and salt stress and its overexpression in tobacco conferred enhanced tolerance to these stresses [113]. Increased tolerance involved enhancement of ABA biosynthesis as well as antioxidant enzymes. In our conditions, drought stress brought about a significantly higher increase in 14-3-3 amount in the Saragolla variety. Information about VGKC in wheat and its possible role in drought tolerance is very limited. However, regulation of K^+^ channels is critical for K^+^ uptake in roots, necessary for osmotic adjustment during the response to drought, as demonstrated in barley [114]. VGKC was up-regulated in the Svevo cultivar and down-regulated in Saragolla. The ATP-binding subunit of Clp P is part of the plastidial Clp protease complex that eliminates misfolded proteins during plastid biogenesis, development [115], or produced in the chloroplast by abiotic stress [115,116]. While information about the role of Clp in protection from stress in non-photosynthetic tissues is lacking, in *Arabidopsis* leaves it has been demonstrated that Clp P confers protection against oxidative stress induced by methyl viologen [117]. Intriguingly, our data showing that levels of Clp P were very strongly increased by drought in both varieties but particularly in Saragolla suggest that Clp P may significantly contribute to protection from drought stress in wheat roots.

## 3. Discussion

The worsening of the global climate change scenario, restricting water availability, imposes a serious threat on durum wheat cultivation, hence the interest in traditional varieties as a source of genetic biodiversity to be exploited for the selection of drought-tolerance traits. In this study the response of Saragolla, a traditional durum wheat cultivar from Southern Italy, less suitable to intensive cultivation but expected to be more tolerant to drought, was investigated in comparison to that of Svevo, an elite widespread-cultivated variety. The response to water stress deficit was investigated in roots, the plant tissue where water scarcity is primarily perceived and a faster response is known to occur. Morphological and physiological parameters were estimated and related to biochemical data such as electrolyte leakage, membrane lipid peroxidation, hydrogen peroxide production and proline synthesis, in order to evaluate the detrimental effect, as well as the response to osmotic stress. Furthermore, comparative qRT-PCR and proteomic analyses were performed in order to shed light on the underlying molecular determinants of the different degree of tolerance showed by the two cultivars: Concerning morphological and physiological parameters, the root-to-shoot ratio and the water content, as well as the RWC of roots and leaves, were examined. It is well ascertained that in many plants during drought the growth of above-ground tissues is inhibited, whereas roots growth continues in an effort to explore a larger soil volume for water uptake [118]. Under our conditions, water stress induced a significantly higher increase in the root-to shoot ratio in the Saragolla cultivar, suggesting that this traditional variety possesses a higher phenotype plasticity to cope with environmental variations, a trait that may have been reduced during the selection of modern elite varieties cultivated under more stable conditions [119]. The RWC, reflecting the metabolic activity of tissues, is a good measure of plant water status and it is widely used as a reliable index for tolerance to drought. Water stress induces a decrease in RWC in a wide variety of plants, a fact that is related to yield loss of crop plants, including wheat [120]. Our analysis demonstrated that the reduction in the RWC of roots of the Saragolla cultivar upon water stress treatment was significantly lower than that of the Svevo cultivar. Measurements of the RWC of leaves, which are more commonly used to evaluate the hydration status of plants, confirmed root results, thereby corroborating the indication that the Saragolla cultivar is less susceptible to dehydration than Svevo. Water deficit stress is strictly associated with the accumulation of ROS, which causes oxidative damage to cellular components, particularly membrane lipids, to the point that the increase in MDA content is considered a suitable marker of membrane deterioration, and low MDA concentrations are associated with drought tolerance in wheat [121]. In the present study, the accumulation of H_2_O_2_, the most stable ROS species, and of MDA, were measured. Both cultivars accumulated H_2_O_2_, with a maximum after two days of water deficit, but Svevo to a significantly higher extent than Saragolla. This trend was paralleled by that of MDA production, which increased in both cultivars during treatment, but to a higher extent in the Svevo one. MDA overproduction, as a consequence of lipid peroxidation, is a measure of membrane damage. Ion leakage determinations confirmed that membrane functionality was impaired upon water stress treatment and that the insult was stronger for the Svevo cultivar. Plant are able of osmotic compensation in response to water stress and proline is one of the more commonly biosynthesized osmolytes, particularly in cereals. Increasing evidence indicates that this compound is also involved in ROS homeostasis, membrane protection and signaling during water stress. Proline accumulation occurred in roots of both cultivars during water deficit but the increase was higher in the Saragolla cultivar. Hence, it is conceivable that proline is a factor contributing to the more efficient ROS management observed in the Saragolla cultivar. Moreover, the higher osmotic compensation is in line with the reduced water loss observed in the Saragolla cultivar. Overall, from morphological, physiological and biochemical pieces of evidence it can be inferred that (i) Saragolla roots are more tolerant than Svevo roots to water stress (ii) the higher root tolerance of Saragolla is a determinant of the higher tolerance of the whole plant (iii) the higher root tolerance of Saragolla appears related to a better ability of counteracting ROS overproduction. Concerning molecular investigations, qRT-PCR experiments were focused on the transcription analysis of genes belonging to the main classes of TFs known to be involved in the expression of tolerance to drought, especially in wheat, such as *WRKY*s, *DREBs* and *DHNs.* In fact, TFs are key regulators of multiple drought-responsive genes, potentially exploitable for improving tolerance [122]. The results demonstrated that an up-regulation of all of the different classes of TFs after 5 days of water deficit occurred in both varieties. This up-regulation was generally higher in the Saragolla cultivar, particularly concerning *WRKY12*, *WRKY34* and *DREB2* genes. Although the involvement of *WRKY*s in the response to water stress is also well ascertained in wheat, information about the involved pathways is quite limited. More information that can be tentatively linked to the physiological and biochemical determinants of tolerance observed under our conditions, is available for *DRBEs*. In fact, it has been shown that DRBEs overexpression in transgenic plants up-regulated water transport proteins such as aquaporins [123], thus contributing to maintaining a healthy water status under stress conditions and that *DREBs* overexpression was associated with an improved root-to-shoot ratio [124], as well as to increased content of antioxidant enzymes and molecules [125]. Proteomic analysis allowed us to ascertain that 5 days of osmotic stress treatment altered the expression of a number of proteins belonging to the abiotic stress, oxidative stress, energy/carbon metabolism and amino acid/nitrogen metabolism functional classes in both cultivars. In particular, in the Saragolla cultivar, a stronger expression of antioxidant enzymes, such as GST, SOD, QR and IR took place. The same trend was observed also for different abiotic stress related proteins, among which, interestingly, DHN and APX-35 appears, from evidence in the literature, to be involved in ROS homeostasis. As for the carbon metabolism/energy production class, it is worth mentioning the strong up-regulation in both cultivars, but particularly in Saragolla, of MDH, a pivotal enzyme for the maintenance of cellular energy homeostasis, whose levels have been shown to increase in drought-tolerant wheat genotypes [82]. In the nitrogen and amino acid metabolism class, it is worth noting the up-regulation of Gsr1 and Gs1a, two isoforms of GS, a key enzyme for amino acid biosynthesis, whose increase can be related to the increased osmolytes biosynthesis, such as proline, during water stress. Gs1a was strongly up-regulated in the Svevo variety and to a lesser extent in the Saragolla variety, while Gsr1 was up-regulated exclusively in the Svevo cultivar. Considering that the induction of the other enzymes involved in nitrogen assimilation and amino acid metabolism was, overall, higher in the Svevo variety, a positive correlation with the higher proline content of stressed Saragolla roots is not straightforward. Finally, the very high up-regulation of the ATP-binding subunit of Clp P, observed in both varieties but especially in Saragolla, suggests that the role of this plastidial protease complex in protecting roots from abiotic stress is worth further investigation. In conclusion, physiological and biochemical analyses demonstrated a higher ability of the Saragolla cultivar to maintain hydration and fitness, as well as ROS homeostasis and membrane integrity, while molecular analyses suggested that the better response of Saragolla to osmotic stress may be related to a higher transcriptional plasticity, which determines a more robust expression of components of the energy production and ROS homeostasis pathways.

## 4. Materials and Methods

### 4.1. Germination, Growth and Water Stress Treatments

Wheat seedlings of the tetraploid *Triticum turgidum ssp durum* cultivars Svevo (Agrisemi Minicozzi, Benevento, Italy) and Saragolla (Syngenta, Milano, Italy) were germinated in Petri dishes and then grown hydroponically in black plastic pots containing half-strength Hoagland’s culture solution in a growth chamber at 24 °C, 50% humidity, under a 16 h light/8 h dark cycle. After seven days of cultivation, half of the seedlings were subjected to osmotic stress treatments by adding in the solution PEG-6000 to a final concentration of 18% (*w/v*), corresponding to an osmotic potential of −1.0 Mpa). Control plants remained in culture solution without any stress-inducing additive. After different times of exposure, the roots and leaves were harvested [126].

### 4.2. Determination of Growth Parameters

Measurement of biomass was carried out at the interval times 0, 2, 5 and 10 days under water stress or normal conditions. All the fresh plant organs were placed in an oven at a temperature of 105 °C for one hour and then were kept at 80 °C until a constant dry weight was obtained. Fresh and dry root and leaf weights were determined by electronic balance. Retained Water was calculated as follows: RW = fresh weight − dry weight, while Relative Water Content: RWC % = (fresh weight − dry weight)/(turgid weight − dry weight) × 100 [127]. The root-to-shoot ratio was determined as the ratio of root dry weight to aboveground dry weight [128].

### 4.3. Determination of Electrolyte Leakage

Two hundred mg of roots from Saragolla and Svevo cultivars were cut into 5 mm slices and submerged in 30 mL of deionized water for 2 h at 25 °C. After incubation, the electrical conductivity was measured by a conductometer (AD31, Adwa, Szeged, Hungary); this parameter was reported as relative electrical conductivity (REC%) values. Boiled samples were used to determine the maximum percentage of electrolyte leakage, which was then calculated using the following formula: REC% = C1/C2 × 100 (C1 = conductivity at 25 °C; C2 = conductivity at 100 °C) [129].

### 4.4. Membrane-Lipid Peroxidation Assay

Membrane-lipid peroxidation was estimated by the MDA assay [130]. One hundred mg of roots from Saragolla and Svevo wheat cultivars were homogenized in liquid N_2_, suspended in 500 µL of 0.1% trichloroacetic acid (TCA), and centrifuged at 15,000× *g*, for 10 min, at 4 °C. A hundred µL of the supernatant was added to 1.5 mL of 0.5% thiobarbituric acid (TBA) in 20% TCA, and incubated for 25 min, at 95 °C. After incubation, the reaction was blocked by placing the samples in ice. After cooling at 25 °C, sample absorbance was measured at 532 nm and 600 nm. The absorbance values measured at 600 nm were subtracted from those measured at 532 nm, and MDA concentration values were calculated by interpolation with a calibration curve obtained with known amounts of MDA.

### 4.5. H_2_O_2_ Production Assay

The concentration of H_2_O_2_ released in the solution was determined with the FOX1 method [131], which is based on the hydrogen peroxide-mediated oxidation of Fe^2+,^ followed by the reaction of Fe^3+^ with xylenol orange dye. Ten roots (1 cm long) were cut and incubated in deionized water for 30 min. The incubation medium was then added to an equal volume of the assay reagent containing 500 mM ammonium ferrous sulfate, 50 mM H_2_SO_4_, 200 mM xylenol orange, and 200 mM sorbitol. After 45 min of incubation, the absorbance of the Fe^3+^-xylenol orange complex was measured at 560 nm, and the H_2_O_2_ concentration was calculated by interpolation from a standard curve obtained with H_2_O_2_ solutions of known concentration.

### 4.6. Determination of Free Proline Synthesis

Five hundred mg of roots from Saragolla and Svevo wheat were powdered in liquid N_2_ and homogenized in 1 mL of 70% ethanol. After centrifugation at 14,000× *g* for 20 min, 0.5 mL of the sample was added to 1 mL of a solution containing 1% ninhydrin, 60% acetic acid, 20% ethanol, and incubated at 95 °C, for 20 min. The absorbance of samples was measured at 520 nm and the amount of free proline was calculated by interpolation from a standard curve obtained with proline solutions of known concentration [132].

### 4.7. qRT-PCR Analysis of Genes Expression

The “Spectrum Plant Total RNA Kit” (Sigma-Aldrich, Milan, Italy) was used to extract the RNA from 5 days stressed or control wheat root samples. RNeasy/QIamp columns and RNase-Free DNase set (Quiagen, Milan, Italy) were used to degrade genomic DNA and obtain an eluate of pure RNA. The extracted RNA was reverse-transcribed to cDNA using “ImProm-II Reverse Transcription System Kit” (Promega, Milan, Italy) and the “Mini thermal cycler BioRad” (Segrate MI, Italy) and stored at −20 °C. Gene specific primers (as shown in Table 3) were designed using the NCBI Primer Blast tool. “EvaGreen 2X qPCR MasterMix-R” kit (Applied Biological Materials, Vancouver, Canada) was used for RT-qPCR. The thermal cycler used “7300 Real-Time PCR System” was set to perform an initial denaturation at 95 °C for 1 min, an annealing phase of 5 min at 95 °C and 40 successive cycles of denaturation (95 °C for 30 s), annealing (60 °C for 30 s) and extension (72 °C for 30 s). Experiments were carried out in triplicate and the relative quantification in gene expression was determined using the 2^−∆∆Ct^ method [133].

### 4.8. Protein Extraction, Two-Dimensional Gel Electrophoresis (2-DE), In-Gel Digestion, and MS Analysis

Protein samples were dissolved in IEF buffer containing, 9 M urea, 4% *w/v* CHAPS, 0.5% *v/v* Triton X-100, 20 mM DTT, 1% *w/v* carrier ampholytes pH 3–10. IPG strips (17 cm pH 4–7, ReadyStrip; Bio-Rad, Hercules, CA, USA) were rehydrated overnight with IEF buffer containing 300 μg of total proteins [134]. Proteins were focused using a ProteanIEF Cell (Bio-Rad) at 12 °C, by applying the following voltages: 250 V (90 min), 500 V (90 min), 1000 V (180 min) and 8000 V for a total of 52 kVh. Furthermore, IPG strips were incubated with 1% *w/v* DTT in 10 mL of equilibration buffer (50 mM Tris–HCl pH 8.8, 6 M urea, 30% *w/v* glycerol, 2% *w/v* SDS and a dash of bromophenol blue) for 15 min, and alkylated with 2.5% *w/v* iodoacetamide in 10 mL of equilibration buffer for 15 min. SDS–PAGE was carried out on 12% polyacrylamide gels (180 × 240 × 1 mm) using electrophoresis buffer (25 mM Tris–HCl pH 8.3, 192 mM glycine and 1% *w/v* SDS), with 120 V applied for 12 h, until the dye front reached the bottom of the gel. 2-DE gels were stained with colloidal Coomassie G250. The images were analyzed for the detection, matching and quantification of protein spots, using PD Quest software (version 8.0.1 (BioRad)), according to the manufacturer’s procedures. Manual inspection of the spots was performed to verify the accuracy of automated gel matching; any errors in the automatic procedure were corrected before quantitative analysis. After normalization of the spot densities against the whole gel densities, the percentage volume of each spot was averaged for 12 gels (three replicates of four different biological samples) and compared between groups (control and PEG-6000 treated) to find out statistically significant (Student’s *t*-test, *p* < 0.05) differences. A two-fold change in normalized spot densities was considered indicative of a differentially expressed protein. Thus, values ≥2 or ≤0.5 were associated with increased or decreased protein amounts after treatment, respectively. Stained protein spots were automatically excised from the gel and de-stained [135] by two washes at 37 °C for 30 min with 100 μL of 100 mM ammonium bicarbonate NH_4_HCO_3_/50% (*v/v*) acetonitrile (ACN). Gel spots were washed twice in 20 μL of 25 mM NH_4_HCO_3_ and then dehydrated with 20 μL of 25 mM NH_4_HCO_3_/50% (*v/v*) ACN, followed by a wash with 20 μL of ACN. Gel pieces were fully dried in a SpeedVacum system (USA). For digestion, the gel pieces were rehydrated in 20 μL of 25 mM NH_4_HCO_3_ solution containing 12.5 g·μL^−1^ trypsin (sequencing grade; Promega, Madison, WI, USA) and incubated on ice for 45 min. The supernatant was discarded, 10 μL of 25 mM NH_4_HCO_3_ were added to the gel, and then it was heated for 2 × 5 min in a microwave oven at 200 W. The peptides were extracted in 0.5 μL of 10% trifluoroacetic acid, with frequent vortexing, for 15 min. Samples were evaporated to dryness and stored at 4 °C until LC-MS-MS analysis. Samples were reconstituted into 20 μL of loading buffer (2% ACN (*v/v*), 0.5% (*v/v*) formic acid) and then were concentrated and desalted on a C18 trap column (PepMap C18; Dionex, Sunnyvale, CA, USA) using a Tempo 1D nanoLC system (Applied Biosystems/MDS Sciex, Toronto, Canada). Peptide separation was achieved on a reversed phase C18 column (PepMap C18, 75 μm i.d., 15 cm, Dionex, Sunnyvale, CA, USA), using an 18 min linear gradient of 5–35% (*v/v*) ACN, 0.1% (*v/v*) aqueous formic acid. Separated peptides were analyzed on a hybrid triple quadrupole/linear ion trap mass spectrometer (4000 Q TRAP LC–MS/MS System; Applied Biosystems) equipped with a heated desolvation chamber interface set to 150 °C and operated under Analyst 1.4.1 software (Applied Biosystems). Up to five peptides precursor ions detected by a linear ion trap MS scan were first subjected to a high-resolution MS scan to determine charge state and molecular weight. Suitable precursors were then fragmented by Enhanced Product Ion Scans (EPI) (Applied Biosystems). In this scan mode, precursors are selected in Q1, fragmented by collision with nitrogen in the Q2 collision cell, and mass analyzed in the Q3 linear ion trap. The collision energy was dynamically adjusted according to the charge state and Mw of the precursors. The resulting spectra are suitable for the sequence analysis. The average cycle time for this experiment was 3.5 s. Proteins were identified from LC–MS/MS data using a novel approach that employs the recently introduced Paragon algorithm present in the commercial ProteinPilot software (Applied Biosystems; [136]). In short, Paragon derives short sequence tags from all MS/MS spectra in an LC/MS/MS experiment. The density of these sequence tags on a given stretch of protein sequence in the database is then graded using the novel concept of Sequence Temperature Values (STVs). Sequence stretches with a high STV; that is, a high density of sequence tags assigned, are then used for an exhaustive search for possible peptide sequences, including variable modifications, seemingly non enzymatic cleavages and amino acid substitutions. For each spot, the list of detected proteins was then consolidated using the ProGroup algorithm incorporated in the 2.0 version of ProteinPilot software (Applied Biosystems). A protein hit is only reported if it has at least one unique, high scoring peptide sequence (maximum missed cleavage = 2, peptide tolerance = ±0.05 Da) assigned to it that is not implemented in other protein hits. This approach efficiently consolidates a large number of similar protein entries from different species expected in the given scenario, without obscuring valuable information about the existence of potential homologs.

### 4.9. Statistical Analysis

Each physiological statistical analysis for three biological replicates was carried out with IBM SPSS Statistics 21 (SPSS Inc., Chicago, IL, USA), performed by one-way ANOVA for multiple comparisons, followed by the non-paired two-tailed student’s *t*-test for comparisons between two groups, with the minimum level of significance being *p* < 0.05. Statistically significant differences (*p* < 0.05) between means are marked with different letters. Bars are represented as the standard deviation of the mean (SD).

## Figures and Tables

**Figure 1 plants-12-00428-f001:**
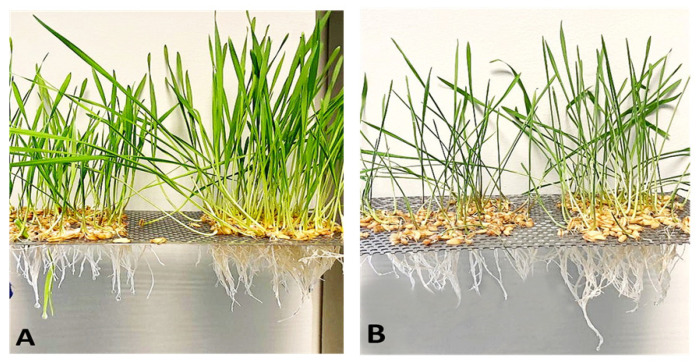
Effect of water stress on the growth of Svevo and Saragolla cultivars. (**A**) Phenotype of seedlings of Svevo (left) and Saragolla (right) seedlings grown for 17 days under control conditions. (**B**) Phenotype of Svevo (left) and Saragolla (right) seedlings grown under control conditions and then subjected to 10 days of osmotic stress by means of 18% PEG-6000 administration, as described in Section 4.1.

**Figure 2 plants-12-00428-f002:**
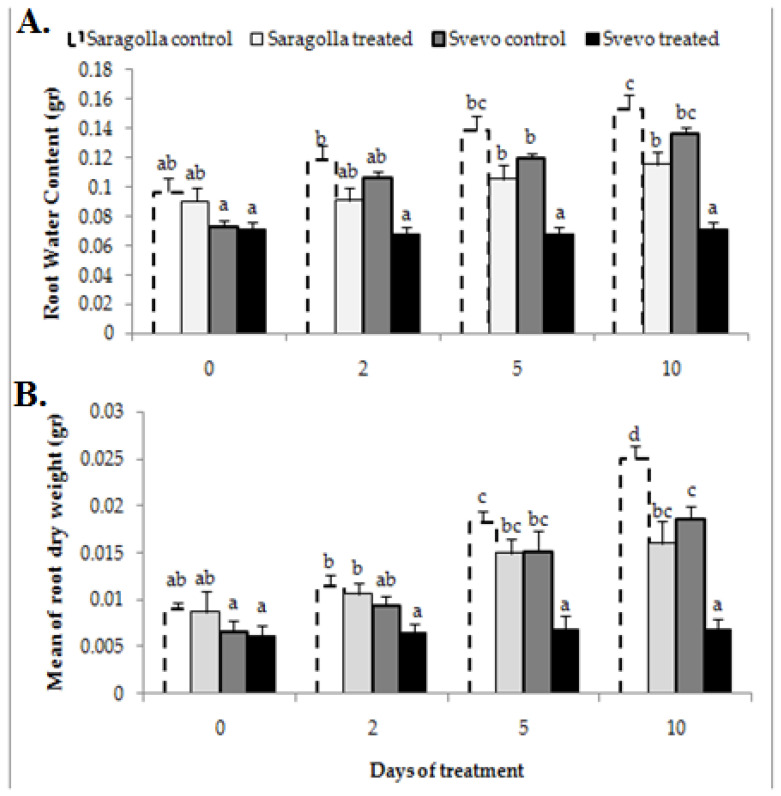
Effect of osmotic stress on water content (**A**) and dry weight (**B**) of roots from Saragolla and Svevo seedlings. Seven days-old Saragolla and Svevo seedlings were subjected to osmotic stress treatment for 10 days by means of 18% PEG-6000 administration. After 2, 5 and 10 days of treatment the water content (**A**) and the dry weight (**B**) of roots were measured as reported in Section 4.2, and values compared to those of control, unstressed seedlings. Results from three independent experiments are reported; values are expressed as the mean ± SD. Statistical significance was attributed by one-way ANOVA test. Bars labeled with dissimilar letters are significantly different (*p* < 0.05).

**Figure 3 plants-12-00428-f003:**
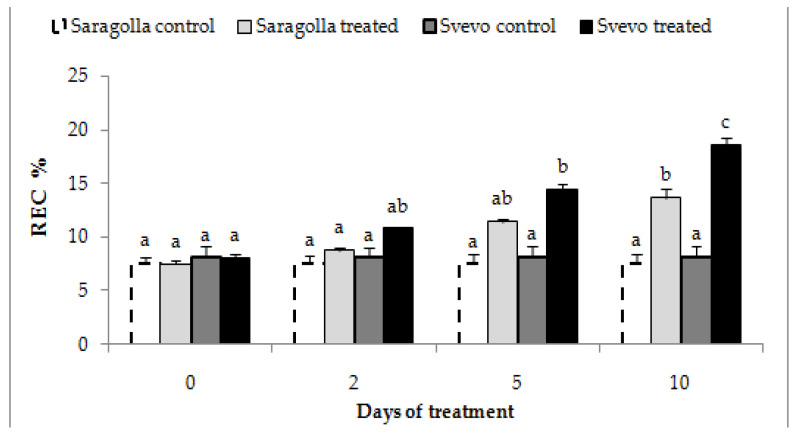
Electrolyte leakage from roots of Saragolla and Svevo seedlings subjected to osmotic stress. Seven days-old Saragolla and Svevo seedlings were subjected to osmotic stress treatment for 10 days, by means of 18% PEG-6000 administration, and after 2, 5 and 10 days the REC% of roots was measured as reported in Section 4.3, and values compared to those of control, unstressed seedlings. Results from three independent experiments are reported; values are expressed as the mean ± SD. Statistical significance was attributed by one-way ANOVA test. Bars labeled with dissimilar letters are significantly different (*p* < 0.05).

**Figure 4 plants-12-00428-f004:**
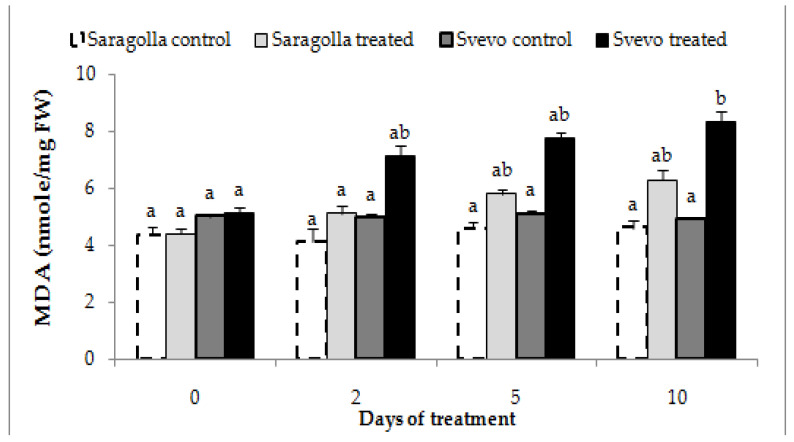
MDA production in roots of Svevo and Saragolla seedlings subjected to osmotic stress. Seven days-old Saragolla and Svevo seedlings were subjected to osmotic stress treatment for 10 days by means of 18% PEG-6000 administration, and after 2, 5 and 10 days, the MDA concentration in roots was measured as reported in Section 4.4, and values compared to those of control, unstressed seedlings. Results from three independent experiments are reported; values are expressed as the mean ± SD. Statistical significance was attributed by one-way ANOVA test. Bars labeled with dissimilar letters are significantly different (*p* < 0.05).

**Figure 5 plants-12-00428-f005:**
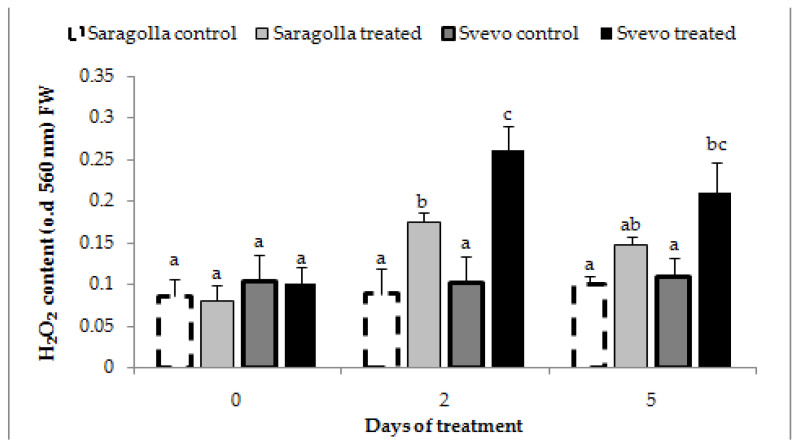
H_2_O_2_ production in Saragolla and Svevo roots subjected to osmotic stress. Seven days-old Saragolla and Svevo seedlings were subjected to osmotic stress treatment for 5 days, by means of 18% PEG-6000 administration and after 2 and 5 days the H_2_O_2_ concentration in roots of control and stressed samples was measured as reported in Section 4.5, and values compared to those of control, unstressed seedlings. Results from three independent experiments are reported; values are expressed as the mean ± SD. Statistical significance was attributed by one-way ANOVA test. Bars labeled with dissimilar letters are significantly different (*p* < 0.05).

**Figure 6 plants-12-00428-f006:**
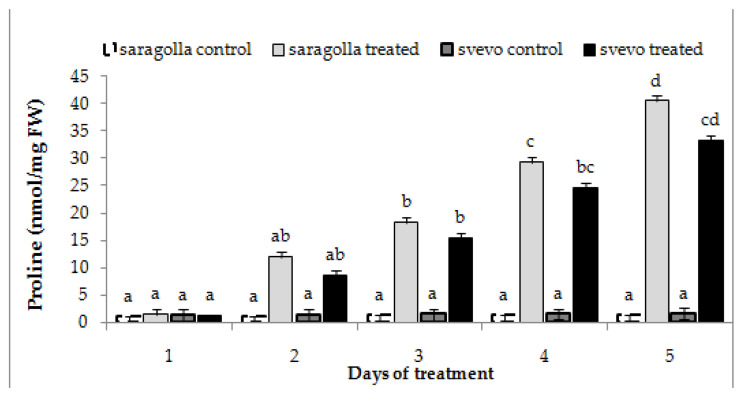
Proline accumulation in Saragolla and Svevo roots subjected to osmotic stress. Seven days-old Saragolla and Svevo seedlings were subjected to osmotic stress for 5 days, by means of 18% PEG-6000 administration, and after 2 and 5 days the proline concentration in roots of control and stressed samples was measured by the acid ninhydrin method, as reported in Section 4.6, and values compared to those of control, unstressed seedlings. Results from three independent experiments are reported; values are expressed as the mean ± SD. Statistical significance was attributed by one-way ANOVA test. Bars labeled with dissimilar letters are significantly different (*p* < 0.05).

**Figure 7 plants-12-00428-f007:**
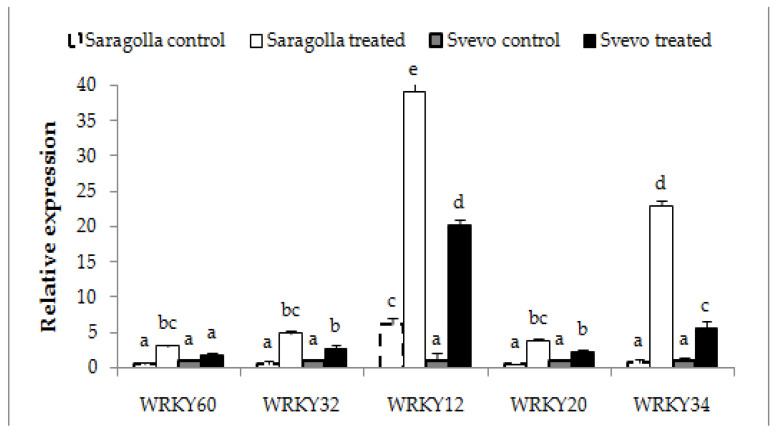
qRT-PCR analysis of *WRKY* genes in Svevo and Saragolla roots subjected to osmotic stress. Seven days-old Saragolla and Svevo seedlings were subjected to osmotic stress treatment for 5 days, by means of 18% PEG-6000 administration. At the end of the treatment, total RNA was extracted and the qRT-PCR analysis of relative expression of *WRKY12*, *WRKY20*, *WRKY32*, *WRKY34* and *WRKY60*, performed as described in Section 4.7. The results of stressed roots are reported as fold changes with respect to non-stressed roots considered as unit. The results represent the mean values ± SD of three independent experiments. Statistical significance was attributed by one-way ANOVA test. Bars labeled with dissimilar letters are significantly different (*p* < 0.05).

**Figure 8 plants-12-00428-f008:**
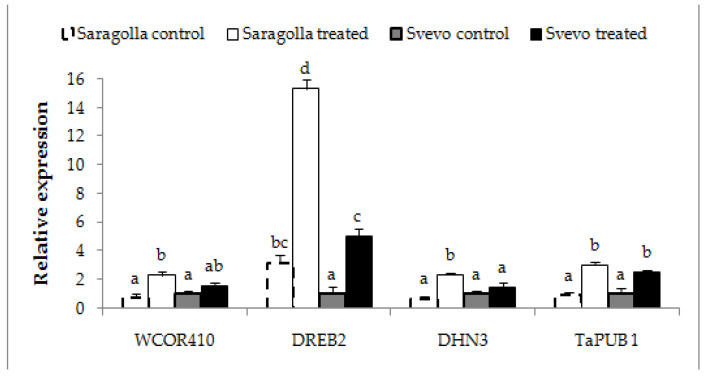
qRT-PCR analysis of *DREB2, DHN3, WCOR410,* and *TaPUB1*genes in Svevo and Saragolla roots subjected to osmotic stress. Seven days-old Saragolla and Svevo seedlings were subjected to osmotic stress treatment for 5 days, by means of 18% PEG-6000 administration. At the end of the treatment, total RNA was extracted and the qRT-PCR analysis of relative expression of *DREB2*, *DHN3*, *WCOR410* and *TaPUB1*, performed as described in Section 4.7. The results represent the mean values ± SD of three independent experiments. Statistical significance was attributed by one-way ANOVA test. Bars labeled with dissimilar letters are significantly different (*p* < 0.05).

**Figure 9 plants-12-00428-f009:**
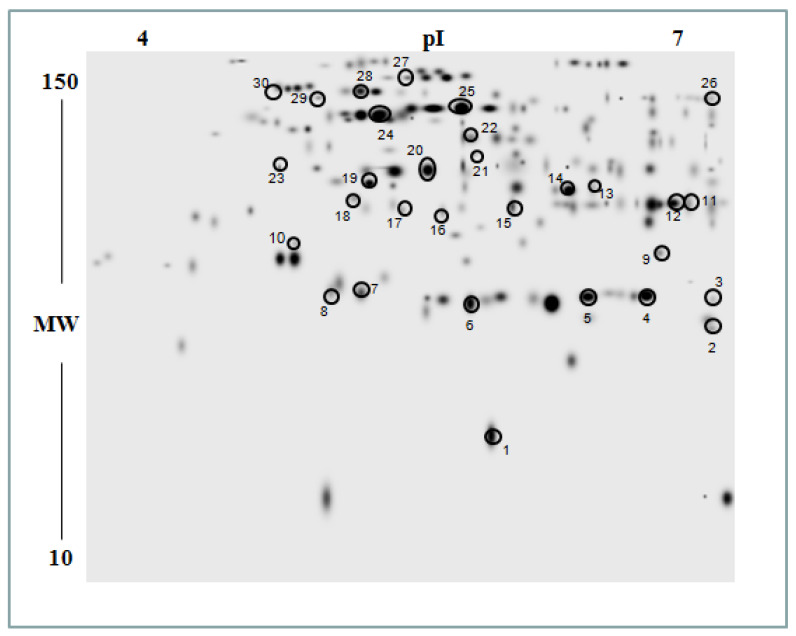
2DE reference map of roots from control or osmotically stressed Svevo and Saragolla seedlings. Protein extracts from roots of control or treated with 18% PEG-6000 for 5 days, Svevo and Saragolla seedlings, were resolved on IPGs (first dimension) and 12% SDS-PAGE (second dimension) and were visualized by colloidal Coomassie blue staining. Gel images were analyzed for the detection, matching and quantification of protein spots, using PD Quest software. Spot numbering coincides with that reported in Table 2; experiments were carried out in triplicate for each biological sample.

**Table 1 plants-12-00428-t001:** Determination of roots and leaves RWC and of root-to-shoot ratio of Svevo and Saragolla seedlings cultivated with full water availability or subjected to osmotic stress by means of 18% PEG-6000 administration for 10 days, as reported in Section 4.2. Results from three independent experiments are reported; values are expressed as the mean ± SD. Statistical significance was attributed by one-way ANOVA test (* *p* < 0.05, ** *p* < 0.01).

Water Condition	Roots RWC %	Leaves RWC %	Root/Shoot Ratio (gr)
Saragolla			
Water control	83 ± 4.5	89 ± 5.7	0.41 ± 0.03
Water drought	62 ± 3.74	74.7 ± 3.92 *	1.15 ± 0.1 **
Svevo			
Water control	76 ± 3.4	84.1 ± 2.98	0.58 ± 0.09
Water drought	40 ± 5.54 *	42.8 ± 4.6 **	0.88 ± 0.12

**Table 2 plants-12-00428-t002:** Differentially represented proteins in roots of Svevo and Saragolla seedlings subjected to osmotic stress. Protein extracts from roots of control or treated with 18% PEG-6000 for 5 days; Svevo and Saragolla seedlings were separated by 2DE and analyzed by LC/MS/MS nano LC-ESI-LIT-MS/MS, as described in Section 4.8. Spot number, protein name, NCBI entry, together with the corresponding fold change between control and PEG-6000 treated samples in the two varieties, are reported. Values in black: Up-regulated proteins. Values in blue: Down-regulated proteins. Values in red: non-differential proteins. Statistical significance was attributed by Student’s test (*p* < 0.05).

*Spot*	*Protein Name*	*NCBI Entry*	*Fold Change* *Svevo Treated/* *Svevo Control*	*Fold Change* *Saragolla Treated/* *Saragolla Control*
Abiotic stress				
**9**	**Dehydrin**	540360825	2.72	3.10
**24**	**Annexin P-35**	1000394	2.32	4.44
**30**	**Hsp70**	379645201	3.77	3.04
**32**	**Hsp70**	379645201	3.82	7.68
Oxidative stress				
**1**	**Glutathione Transferase F3**	23504741	1.32	3.27
**2**	**Quinone Reductase**	62363163	1.24	3.65
**5**	**Superoxide Dismutase**	226897531	1.88	2.17
**11**	**Peroxidase**	732974	5.46	4.03
**12**	**Peroxidase**	732974	2.05	4.02
**13**	**Isoflavone Reductase**	2123653802	2.71	10.1
Energy and carbon metabolism				
**3**	**Malate Dehydrogenase**	1318868161	6.04	35.02
**15**	**UDP-glucose 6-dehydrogenase 3**	108711178	6.68	0.2
**16**	**Succinate-CoA Ligase β chain**	75261432	7.42	1.49
**17**	**Adenylosuccinate synthetase**	1616659	4.08	2.63
**18**	**Enolase**	461744078	6.74	6.9
**19**	**Phytoene synthase 1**	154550145	4.89	4.1
**20**	**3-Phosphoglycerate kinase**	28172911	4.49	2.41
**25**	**Phosphoglycerate mutase**	144952816	4.78	0.51
**29**	**Aconitate hydratase**	474314661	2.97	3.07
Amino acid and nitrogen metabolism				
**6**	**Cysteine synthase**	543181	4.11	0.46
**7**	**Glutamine synthetase Gsr1**	40317416	4.73	0.85
**10**	**Glutamine synthetase Gs1a**	321531577	10.52	2.78
**14**	**Ferredoxin nitrite reductase precursor**	218963622	1.5	3.37
**22**	**Ferredoxin nitrite reductase precursor**	218963622	4.93	3.72
**23**	**S-Adenosylmethionine synthetase**	166235928	8.52	5.21
**27**	**Putative Glycine decarboxylase subunit**	22204118	7.06	0.93
**28**	**Ferredoxin nitrite reductase precursor**	218963622	1.2	5.49
Miscellaneous				
**4**	**Probable voltage-gated potassium channel subunit β**	2123890642	2.72	0.44
**8**	**14-3-3 protein**	40781605	3.16	5.63
**31**	**ATP-Dependent Clp protease ATP-binding subunit**	2123891895	9.39	19.06

**Table 3 plants-12-00428-t003:** Primers sequences designed for quantitative real time PCR analysis of genes in wheat.

Genes	Forward Primer	Reverse Primer
WRKY12	ACGGCCAGAAGCCCATCAA	GGTGCTCGCCCTCGTAAGTC
WRKY20	CAACCAGGACCCAGCAAAGA	CTCCATCTTGACTGGGGCAT
WRKY32	CAAGCGCATCCGGGAGGAGT	CGTTCCGCTTCTGGCTGTGC
WRKY34	CGAGGTGGACGAGCCAGGT	CGGTGAGACTGAGGTGTTGTT
WRKY60	ACCAGCCCTTCAGGACCAA	CTGCCAAGAACCACGAGACA
WCOR	ATGGAGGATGAGAGGAG	GCTTGTCCTTGATCTTG
DHN	ATGGAGCACCAGGGGC	GCAGCTTGTCCTTGATCTTG
DREB	AAGAAAACAGGCGACAAGAT	ACGAAGCACAAAAAACTA
TaPUB1	AAATCTCCAGTCATCCACTTC	CCATCTTCATTACCTTGCCATAC

## Data Availability

The data presented in this article are available on request from the corresponding authors.

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
