# Peer review of "Comparative Analysis of the Response to Polyethylene Glycol-Simulated Drought Stress in Roots from Seedlings of “Modern” and “Ancient” Wheat Varieties"

_plants, 2023, doi:10.3390/plants12030428_

Round 1

Reviewer 1 Report

The title is not properly designed, instead of using ‘Water stress’, the authors should simply use drought. This raises another question, the drought stress is not from real soil drought treatment, instead, it is from the artificial PEG6000 treatment.

The language needs to have more work, generally, it reads too lengthy. The authors may need to consider making it shorter.

Line 9 to line 11, reads odd.

Line 15, not necessary to have the words ‘that is the most common abiotic stress experienced by plants during their lifecycle’.

The introduction is not well designed, since the authors highlight the root traits as the significant components of the study, the introduction should focus more on the physiological responses from the roots. Apparently, not much information regarding the root was indicated in the introduction part!

Line 105, it is not very right to say ‘water stress. It should be drought stress or osmotic stress. Besides, what is the concentration of PEG as the treatment? Is it 18% ? or 18% PEG6000 was added to the irrigation solution? What is the ratio and the recipe? This is totally missing.

In Figure1.An obvious growth biomass difference between left and right. This is not scientifically right since under control conditions there should be zero differences between your two varieties. It is hard to tell if Saragolla is more tolerant than Svevo because 1) the root density (Fig.1 A), the right side has more roots than the left side; 2) seedlings from the right side have significantly higher biomass.

Line138, by adding how much 18% PEG-6000?

Line 139, why don’t the authors add A and B, instead of using an upper panel and lower panel?

Line 619, grammatically is not right.

Reviewer 2 Report

 In this manuscript entitled “Comparative Analysis of the Response to Water Stress of Roots from Seedlings of “Modern” and “Ancient” Wheat Varieties” Licaj et al. have performed comparative analysis of the two wheat varieties under water stress. The authors have analyzed various biochemical parameters such as hydrogen peroxide production, electrolyte leakage, membrane lipid peroxidation, proline synthesis under this condition. The authors also performed qRT-PCR analysis of the various important genes that are known to involve in abiotic stress. Further, the authors have identified differentially expressed proteins under control and water stress conditions in these two varieties using 2D gel analysis. Overall, the study is clearly designed and presented well with quality figures and the conclusions are well supported by the results. 

Specific comments/suggestions

Line 725: Change to “after different times of exposure”

Line 730:  Change to “water stress”

 Line 749: Check the sentence

Line 750: Was the kit used for the total RNA isolation or mRNA extraction?

Line 795: Change as “reverse-transcribed”

Line 797: Double check the volume of the RNA and the Oligo (dt) primers

Section 4.7: Cut short the procedure of the total RNA isolation

Round 2

Reviewer 1 Report

The primary concern was regarding the plant materials utilised in this research. Since the other publications using the same varieties for drought studies have been acknowledged, it seems acceptable. 

I, therefore, don't have any further comments.